# A Comparative Study on Cognitive Assessment in Cerebellar and Supratentorial Stroke

**DOI:** 10.3390/brainsci14070676

**Published:** 2024-07-03

**Authors:** Qi Liu, Yumei Zhang

**Affiliations:** 1Department of Neurology, Beijing Tiantan Hospital, Capital Medical University, Beijing 100070, China; 2Department of Rehabilitation, Beijing Tiantan Hospital, Capital Medical University, Beijing 100070, China; 3China National Clinical Research Center for Neurological Diseases, Beijing Tiantan Hospital, Capital Medical University, Beijing 100070, China

**Keywords:** cerebellum, cognitive functions, stroke, comparison study, supratentorial infarction, Cerebellar Cognitive Affective Syndrome

## Abstract

This study aims to understand the cognitive profiles of cerebellar infarction patients and compare them to those with supratentorial infarctions, particularly frontal infarctions. This current study also aims to find reliable assessment tools for detecting cognitive impairment in cerebellar infarction patients. A total of fifty cerebellar infarction patients, sixty supratentorial infarction patients, and thirty-nine healthy controls were recruited. The Mini-Mental State Examination (MMSE), Montreal Cognitive Assessment (MoCA), Addenbrooke’s Cognitive Examination III (ACE-III), and Cerebellar Cognitive Affective Syndrome scale (CCAS-s) were used to assess global cognitive function. An extensive neuropsychological assessment battery was also tested to evaluate the characteristics of each cognitive domain. To assess the features of cognitive function, a comprehensive neuropsychological evaluation tool was also utilized. The cerebral infarction patients demonstrated cognitive impairment comparable to those with frontal infarcts, notably characterized by disturbance in attention and executive function. However, the degree of cognitive impairment was comparatively milder in cerebellar infarction patients. Furthermore, the patients in the cerebellar group had worse scores in the ACE-III and CCAS-s compared to healthy controls. The two assessments also demonstrated a significant area under the curve values, indicating their effectiveness in distinguishing cognitive impairment in cerebellar infarctions. In conclusion, cognitive impairment in a cerebellar infarction resembles frontal lobe dysfunction but is generally mild. It can be accurately assessed using the ACE-III and CCAS-s scales.

## 1. Introduction

The role of the human cerebellum extends beyond facilitating purposeful movement and its significant contribution to cognition is gaining widespread recognition [1]. Patients who suffer cerebellar disorders have been explored to present the disturbance with executive function, visuospatial cognition, language, and emotion–affect, leading to the definition of Cerebellar Cognitive Affective Syndrome (CCAS) [1,2]. Previous studies among patients with cerebellar damage showed that cognitive dysfunction patterns differ with patients with supratentorial lesions [3]. In general, patients with CCAS did not display significant performance of memory loss as shown in cortical dementias; the cerebellar cognitive impairment profile was “frontal-like”, which is characteristic of the damage in attention and executive function [4]. Executive impairment has been observed in multiple research studies about cerebellar disorders, impacting attention [5,6], integration and organization [7], sequencing [8,9], and task planning [5,10], as well as inhibition of inappropriate responses [11]. And, compared with supratentorial lesions, cerebellar damage always leads to more subtle cognitive impairments [12].

The current discourse surrounding cerebellar cognition has shifted from questioning its presence to exploring the manner in which it contributes to cognitive processes [13]. It has been suggested that the cerebellar neural circuits connecting to the prefrontal, posterior parietal, and limbic cortices are responsible for the cerebellar involvement in cognitive processing [13]. Evidence from a diffusion-weighted imaging study utilizing a probabilistic tractography approach revealed abnormal modifications in the precuneus, cingulate gyrus, and frontal-temporal cortex, demonstrating a significant contribution to cognitive deficits, specifically executive functions, subsequent to cerebellar diseases [14]. The alterations in the cerebro–cerebellar functional connectivity are also implicated in executive function networks across various disorders such as dementia [15], Parkinson’s disease [16], mental illness [17,18,19,20], substance use disorder [21], and heart disease [22,23,24]. In addition, the theory of “dysmetria of thought” (DoT) has been posited as a key conceptual model to elucidate cognitive deficits in cerebellar disorders [25]. This hypothesis highlights the cerebellum’s critical role in regulating and modulating prefrontal discharges rather than their creation, providing an explanation for the mild or short-term cognitive deficits following cerebellar damage.

The significance of cerebral–cerebellar connections in cognitive functioning has been established [26]. It is hypothesized that cognitive deficits resulting from cerebellar lesions may share certain similarities with those arising from supratentorial lesions. However, the predominant focus of most studies has been on the characteristics of cognitive impairments in cerebellum disorders, typically utilizing healthy control groups, while disregarding comparisons with patients suffering supratentorial lesions [8,12,27]. Moreover, certain studies encompassed individuals with diverse cerebellar disorders, and it is worth noting that the variability of the disease may exert an influence on the outcomes [4,8]. Cerebral infarction, characterized by a confined lesion without the presence of additional brain abnormalities like atrophy or hydrocephalus, presents a suitable disease model for examining functional abnormalities in distinct affected regions. We aimed to investigate the cognitive profiles in isolated cerebellar infarction patients and compare them to those who experienced supratentorial infarctions, especially frontal infarctions. Furthermore, we attempted to identify an appropriate cognitive screening tool for detecting cognitive deficits in individuals who suffered localized cerebellar lesions.

## 2. Materials and Methods

### 2.1. Participants

The participants were recruited consecutively from Beijing Tiantan Hospital and surrounding residential communities between December 2022 and September 2023, and then divided into three distinct groups: (1) individuals diagnosed with focal cerebellar infarctions; (2) individuals diagnosed with focal supratentorial brain infarctions; and (3) healthy participants. The radiologist identified the lesion locations for the cerebellar and supratentorial infarction groups using T2-fluid attenuated inversion recovery (FLAIR) sequences on the 3.0 T magnetic resonance imaging scanner (SIEMENS, Germany). The inclusion criteria for the patient groups are initial occurrence of minor stroke, as defined by a National Institutes of Health Stroke Scale (NIHSS) score of ≤3 upon admission, which is defined as a National Institutes of Health Stroke Scale (NIHSS) score of ≤3 upon admission, and being within the acute phase of 14 days [28]. The exclusion criteria for individuals are having other brain disorders, psychiatric conditions, and taking psychotropic drugs that affect cognitive functioning. We also specifically chose patients with frontal lobe infarcts from the supratentorial group of patients in order to compare the cognitive profile between patients in the frontal and cerebellar groups for sensitivity analyses. For the healthy group, all participants underwent medical examinations to confirm they met the recruitment criteria. And, they matched the age and education of the patients group. Prior to participating, all individuals provided written informed consent for the procedure, which had obtained approval from the ethics committee of Beijing Tiantan Hospital.

This study was conducted in compliance with the ethical principles outlined in the Declaration of Helsinki.

### 2.2. Demographics and Clinical Data Collection

We collected demographic and clinical data, including age, sex, level of education, medical histories of diabetes mellitus, dyslipidemia, hypertension, and the habit of smoking and drinking. The information on the stroke onset consisted of onset duration and the size of the infarction. Motor functions were assessed in all participants. The neurological deficit was measured using the NIHSS score. The ataxia severity was evaluated using the International Cooperative Ataxia Rating Scale (ICARS), while the balance dysfunction was assessed using the Brunel Balance Assessment (BBA) [29,30]. Additionally, participants’ levels of anxiety and depression were measured using the Self-rating Anxiety Scale (SAS) and Self-rating Depression Scale (SDS), respectively [31,32].

### 2.3. Cognitive Data Collection

The neuropsychological assessment was conducted within two weeks of disease onset. The presence of any pre-existing cognitive impairment was excluded by utilizing the Information Questionnaire on Cognitive Decline in the Elderly (IQCODE) [33]. Global cognitive function was assessed using four scales: the Mini-Mental State Examination (MMSE) [34], Montreal Cognitive Assessment (MoCA) [35], Addenbrooke’s Cognitive Examination III (ACE-III) [36], and CCAS scale (CCAS-s) [37]. The CCAS-s, a 10-item neuropsychological test battery, is used to assess CCAS [37]. Each item has a designated cutoff score to determine pass or fail. CCAS is classified as possible with a single failed test, probable with two failed tests, and definite with three or more failed tests. This scale has been translated into Chinese recently [38]. In addition, a series of evaluations was used to further examine five cognitive domains: attention, visuospatial function, language processing, episodic memory, and executive function (for more details, please see Appendix A).

To facilitate comparisons across measures, each cognitive scale performance was transformed into z scores [39]. For each cognitive domain, scores of interest (z, mean/composite) were determined. The definition of cognitive impairment was determined by achieving a score that is one or more standard deviations below the average of the healthy control group [40].

### 2.4. Statistical Analysis

Continuous data were characterized by their mean and standard deviation (SD), or median and quartiles, whereas categorical data were presented as frequency and percentage. The demographics, clinical attributes, and cognitive performance were compared using the Mann–Whitney test for two groups and the Kruskal–Wallis H test for three groups. Pairwise comparisons were conducted using the Dunn–Bonferroni post hoc test. Chi-square analysis was employed for categorical variables. Receiver operating characteristic (ROC) curves were used to assess the discriminatory capacity of various scales in detecting cognitive dysfunction in cases of cerebellar infarctions. We conducted statistical evaluations using the SPSS version 26.0 (IBM, Armonk, NY, USA).

## 3. Results

### 3.1. Demographic and Clinical Characteristics

We recruited 149 patients, constituting 50 patients with cerebellar infarctions, 60 patients with supratentorial infarctions, and 39 healthy controls. Age, sex, educational level, the incidence of the comorbidities, and the performance of SDS showed no significant difference among the three groups (Table 1). Patients with cerebellar and supratentorial infarctions, however, were more anxious, with a more severe SAS score compared with control groups (*p* = 0.001). Cerebellar infarction patients had a longer disease duration compared to patients with supratentorial infarctions (8.7 ± 4.9 vs. 7.1 ± 3.8, *p* = 0.03). No difference was shown between the two patient groups in lesion volume and the NIHSS score. Patients with cerebellar infarctions obtained poor scores on the ICARS and BBA, indicating ataxia and balance problems. Furthermore, in the sensitivity analysis, no significant differences were observed in these variables, except for the SAS score, among the cerebellar infarction group, frontal infarction group, and healthy group (Appendix A).

### 3.2. Global Cognitive Performance among Three Groups

The global cognitive abilities of the three groups are shown in Table 2. There is no significant difference in the performance of MMSE among the three groups (*p* = 0.05). Compared to healthy controls, the MoCA scores of patients with supratentorial infarctions were significantly lower than those of the healthy control group (*p* = 0.002) and patients with cerebellar infarctions (*p* = 0.02), but there was no significant difference between the cerebellar infarction group and the control group (*p* = 1.00). Compared with controls, both the cerebellar and supratentorial infarction groups obtained significantly worse performance in the ACE-III score (*P*_cerebellar infarction group_ = 0.001, *P*_supratentorial infarction group_ < 0.001) and the CCAS-s total score (*P*_cerebellar infarction group_ = 0.02, *P*_supratentorial infarction group_ < 0.001).

According to the MMSE, only one (2.0%) patient was diagnosed with cognitive impairment in the cerebellar group, while 16 (26.7%) in the supratentorial group were diagnosed with cognitive impairment. According to the MoCA, four (8.0%) individuals in the cerebellar infarction group were diagnosed with cognitive impairment, while 25 (41.7%) individuals in the supratentorial infarction group were diagnosed with cognitive impairment. Based on the ACE-III, a substantial rise was observed in the number of individuals identified with cognitive deficits, with 20 (40.0%) cerebellar infarction patients and 31 (51.7%) supratentorial infarction patients. According to the CCAS-s scale, 15 (30.0%), 7 (14.0%), and 15 (30.0%) patients with cerebellar infarctions met the criteria that determined possible, probable, and definite CCAS, respectively.

### 3.3. Discriminative Ability of Different Scales to Differentiate Patients from Cognitively Normal Group

ROC curves were plotted to assess the MMSE, MoCA, ACE-III, and CCAS-s in distinguishing patients from healthy controls. For differentiating patients with cerebellar infarctions from healthy controls, the area under the curve (AUC) with a 95% confidence interval (CI) was 0.72 (0.62–0.83), 0.67 (0.56–0.78), and 0.69 (0.59–0.80) for ACE-III, the CCAS-s total score, and the number of CCAS-s failed items, respectively. However, no significant AUC values were obtained for the MMSE (*p* = 0.99) and MoCA (*p* = 0.32). In contrast, all global screening scales in this study showed significant discriminative abilities for differentiating patients with supratentorial infarctions from healthy controls. The AUC curves of the MMSE, MoCA, ACE-III, CCAS-s total score, and number of CCAS-s failed tests were 0.62 (0.51–0.72), 0.70 (0.60–0.80), 0.80 (0.72–0.89), 0.79 (0.70–0.88), and 0.82 (0.74–0.90). Additionally, we also evaluated that those three scales can distinguish frontal infarction patients from healthy controls: MoCA (AUC = 0.68, 95% CI: 0.56–0.81), ACE-III (AUC = 0.74, 95% CI: 0.62–0.85), CCAS-s total score (AUC = 0.76, 95% CI: 0.65–0.87), and number of failed tests (AUC = 0.79, 95% CI: 0.69–0.89), respectively (Figure 1, Figure 2 and Figure 3 and Appendix A).

According to the Youden index, the optimal cutoff value of ACE-III, the total score of CCAS-s, and the number of CCAS-s failed tests to detect cognitive impairment in patients with cerebellar infarctions, supratentorial infarctions, and frontal infarctions are consistently determined as 88, 91, and 2, respectively. Sensitivity ranges from 94.9% to 100% for ACE-III and CCAS-s in patients with cerebellar infarctions, while specificity ranges from 40.0% to 48.0%. Initial cutoffs show CCAS-s sensitivity for identifying possible/probable/definite states as 74.0%/44.0%/30.0% with specificities of 48.7%/87.2%/97.4%. Furthermore, the ideal cutoff for MMSE in supratentorial infarction patients is 25, with a sensitivity of 100.0% and specificity of 38.3%. Sensitivity and specificity for other scales in patients with supratentorial and frontal infarctions range from 68.4% to 100.0% and 38.3% to 87.2%, respectively (Appendix A).

### 3.4. Comparison of Cognitive Domain Impairment Characteristics among Groups

In comparison to the control group, patients diagnosed with cerebellar infarctions demonstrated attention disturbance (*p* = 0.003) and executive deficits (*p* = 0.002), affecting 22 (44.9%) and 29 (58.0%) individuals, respectively. Furthermore, patients with supratentorial infarctions not only experienced significant attention dysfunction (*p* < 0.001) and executive damage (*p* < 0.001) but also exhibited deficits in episodic memory (*p* = 0.02), affecting 45 (75.0%), 34 (56.7%), and 26 (44.1%) individuals, respectively. Moreover, the supratentorial group exhibited a more pronounced attention impairment (*p* < 0.01) and executive deficit (*p* = 0.02) compared with the cerebellar group (Table 3, Table 4 and Appendix A).

Furthermore, we also calculated the relative odds ratios (ORs) to assess the risk of impaired cognitive domain in each patient group. Patients with cerebellar infarctions had ORs with 95% CI of 1.58 (1.20–2.09) and 2.02 (1.42–2.87) for impairment in attention and executive function, respectively. In contrast, patients with supratentorial infarctions had ORs of 3.49 (2.21–5.49) and 1.95 (1.42–2.69) for attention and executive impairment, respectively. Additionally, the OR for episodic memory impairment in the supratentorial infarction group was 1.51 (1.16–1.97) (Table 3 and Table 4).

### 3.5. Sensitivity Analysis: Comparing Cognitive Function between Patients with Cerebellar Infarctions and Frontal Infarctions

In the supratentorial infarction group, a combined 38 individuals were identified with infarctions in the frontal lobe. A statistical analysis did not uncover any notable differences in demographics and clinical features between the frontal and cerebellar groups (Appendix A). Patients with frontal lobe infarctions exhibited impaired attention (*p* < 0.001) and executive functioning (*p* = 0.004), while their memory function did not show significant impairment (Table 5). This suggests a similar pattern of cognitive impairment as seen in the cerebellar infarction group. Notably, patients with frontal lobe infarctions exhibited significantly greater attention impairment severity than those with cerebellar infarctions (*p* = 0.009) (Table 5 and Appendix A).

## 4. Discussion

In this study, patients with cerebellar infarctions were commonly linked to executive dysfunction, akin to the cognitive impairment observed in frontal lobe infarctions. Nevertheless, they generally lead to less memory decline in comparison to other supratentorial infarctions. The cognitive deficits observed in cerebellar infarction patients are generally mild, with the ACE-III and CCAS-s demonstrating a strong discriminatory capacity for detecting cognitive impairments in this population.

Executive dysfunction is the main characteristic of cognitive impairment in patients with cerebellar infarctions. The likelihood of executive dysfunction in individuals with cerebellar infarctions is twice that observed in healthy subjects. We did not find a significant memory impairment in patients with cerebellar infarctions, which is similar to the cognitive profile of patients with frontal lobe lesions. According to the functional neuropsychological mapping of brain lobes, individuals who have lesions in the frontal lobe are predominantly associated with metacognitive and executive functioning, such as sequencing [8,9], scheduling [5,10], inhibition [11], and organization [7]. Conversely, patients with lesions in the temporal, parietal, and occipital lobes are categorized as the “non-frontal group”, primarily characterized by their involvement in perception, sensory, and associative abilities [41]. Our study confirms that individuals with cerebellar infarctions display cognitive patterns resembling those of the frontal lobe, possibly stemming from a disrupted connection between the cerebellum and prefrontal cortex [42]. Prior research suggests that the cerebro–cerebellar circuit involves input via cortico-ponto-cerebellar projections and output through the cerebello–thalamic-cortical pathways [4,43,44]. Recent neuroimaging research has also revealed the functional connections between the cerebrum and cerebellum: the cerebellum engages the dorsal and ventral attention networks, along with the frontoparietal, default, and salience networks, closely resembling the intrinsic connectivity networks of the cerebral hemispheres [26,45]. The compromised functional integration of the cerebellum with the cerebrum, particularly with the prefrontal association cortex, is often associated with cognitive deficits in cerebellar conditions [14,46].

In this study, we also find that cerebellar damage tends to result in a milder form of cognitive impairment, which is considerably more subtle compared to the cognitive deficits observed in cortical lesions. This may be attributed to the fact that supratentorial lesions primarily induce cognitive abnormalities, while cerebellar lesions primarily disrupt the regulation of cognitive functions [14,26]. These findings further support the DoT theory, which posits that the cerebellum plays a crucial role as a central hub within the distributed neural circuits that facilitate sensorimotor, cognitive, autonomic, and affective functions. According to this theory, the cerebellum not only regulates various aspects of movement such as pace, strength, tempo, and precision but may also influence the velocity, ability, uniformity, and suitability of cognitive functions [47]. Consequently, when there is cerebellum damage, there is a discrepancy between actual reality and perceived reality, resulting in erratic efforts to correct thinking faults [25].

Our study revealed that the ACE-III test displayed a considerable sensitivity for identifying cognitive impairment in individuals who suffered cerebellar infarctions. The AUC of the ACE-III total score in the cerebellar infarction group was 0.72, which was consistent with those in previous studies (0.79 in a heterogeneous focal cerebellar disorders cohort) [4]. The possible reason is that the impact on cognitive abilities that heavily rely on cerebellar function, such as attention, performance, and verbal fluency assessments, are accurately captured in the ACE-III score. The MMSE and MoCA, however, showed no discriminating ability to detect CCAS in cerebellar infarction patients [37]. The ceiling effect of these scales has also been found in detecting cognitive impairment for cerebellar degenerative diseases [48,49]. The probable explanation may be attributed to some tests within the two scales, which are mini versions of the original test design (e.g., digit span backward), and may not possess sufficient sensitivity to accurately assess mental flexibility in patients suffering cerebellar disorders. In addition, the two scales primarily focus on identifying episodic memory dysfunction and may not be appropriate for cerebellar disorders.

In addition, this present study further confirms the excellent discriminatory ability of the CCAS-s in detecting cognitive dysfunction in both the supratentorial and cerebellar groups. The CCAS-s, developed in 2018 using a large cohort of patients with cerebellar diseases, has proven to be valuable in improving the diagnosis of CCAS in clinical practice [37]. Our study found that the AUC of the CCAS-s total score in the cerebellar infarction group was 0.69, consistent with previous studies focusing on Spinocerebellar Ataxia (SCA) and Friedreich’s Ataxia (FRDA), which also demonstrated good specificity and sensitivity in patients with cerebellar disorders [48,50,51,52]. Furthermore, our study revealed that the CCAS-s was equally effective at identifying cognitive impairment in patients with supratentorial infarctions. Unlike other standard screening tests, the CCAS-s has a more comprehensive assessment scale ranging from 0 to 120 points, allowing for a more detailed examination and providing additional quantitative information regarding a patient’s performance in each domain, which can be beneficial for longitudinal follow-up. The CCAS-s can not only be applied to cerebellar disorders but also utilized for screening and evaluating post-stroke cognitive impairment in future studies.

This study aimed to investigate the impact of cerebellar involvement on cognitive function. Notably, it is the first study to evaluate and compare cognitive patterns between cerebellar and supratentorial brain region damage in patients with cerebral infarctions. The utilization of independent groups of patients with confirmed focal cerebellar lesions, which are etiologically homogenous, enhances the credibility of the results. Additionally, our study only included stroke patients in the acute phase, which may exclude to some extent the effects of structural remodeling and compensation of supratentorial brain regions on cognitive damage produced by cerebellar infarctions. We also indicated that the ACE-III and CCAS-s cognitive screening tools are effective in detecting cognitive dysfunctions associated with focal cerebellar damage.

However, our study has several limitations. Firstly, it is noteworthy that the occurrence of dizziness among patients with acute cerebellar infarctions may have influenced the cognitive evaluation. In addition, given the established link between emotion processing and cognition, it is pertinent to mention that the patients displayed heightened levels of anxiety in comparison to the control group, potentially resulting in an overestimation of the prevalence of cognitive impairment. Furthermore, this study did not take into account potential confounding factors that may affect cognitive function, such as the etiology of the stroke and fatigue [53,54]. This oversight may cause a certain degree of bias. It is also important to highlight that our evaluation of cognitive domains, specifically visuospatial ability and language function, was limited to a few scales, which may have resulted in an imprecise assessment of these domains. Additionally, this current study serves as a preliminary investigation, indicating that the absence of an a priori sample size calculation could potentially compromise the statistical efficacy of the research. And, this study’s sample size is relatively limited, highlighting the need for prudence when extrapolating the findings. Despite these limitations, we are confident that our results offer valuable perspectives and can lay the groundwork for future inquiries in this field, with more precise sample size calculations. In the future, longitudinal clinical and neuroimaging studies are needed to explore the cognitive trajectory of cerebellar and supratentorial infarction patients and further discover the long-term brain network mechanism of the effect of cerebro–cerebellar loops on cognitive function.

## 5. Conclusions

This study identified the attention damage and executive dysfunction in cerebellar infarctions, which is similar to the cognitive impairment observed in frontal lobe infarctions. This study aims to provide a neuropsychological perspective on the distinct role of the cerebellum in cognitive impairment by comparing cognitive abnormalities in cerebellar infarcts with those in supratentorial infarcts. Future research should focus on conducting more structured and functional connectivity studies to further investigate the impact of cerebro–cerebellar loops on cognitive function.

## Figures and Tables

**Figure 1 brainsci-14-00676-f001:**
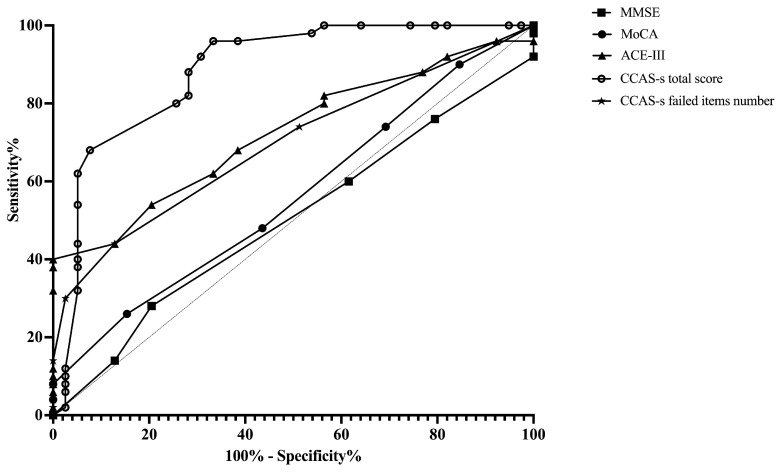
Discriminative ability of MMSE, MoCA, ACE-III, and CCAS-s to identify cognitive impairment in patients with cerebellar infarctions. Abbreviations: MMSE: Mini-Mental State Examination; MoCA: Montreal Cognitive Assessment; ACE-III: Addenbrooke’s Cognitive Examination III; and CCAS-s: Cerebellar Cognitive Affective Syndrome scale.

**Figure 2 brainsci-14-00676-f002:**
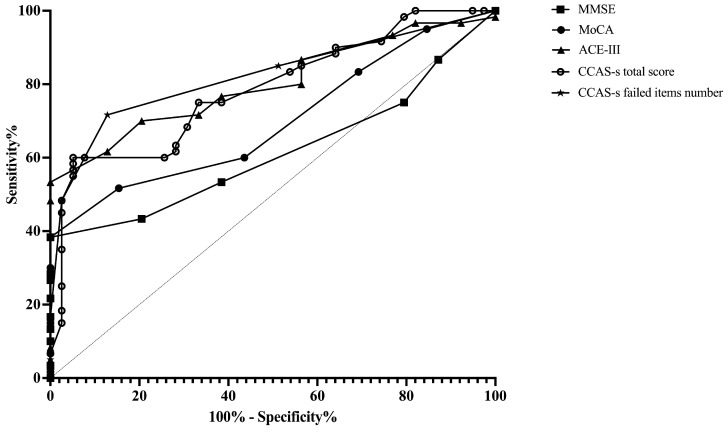
Discriminative ability of MMSE, MoCA, ACE-III, and CCAS-s to identify cognitive impairment in patients with supratentorial infarctions. Abbreviations: MMSE: Mini-Mental State Examination; MoCA: Montreal Cognitive Assessment; ACE-III: Addenbrooke’s Cognitive Examination III; and CCAS-s: Cerebellar Cognitive Affective Syndrome scale.

**Figure 3 brainsci-14-00676-f003:**
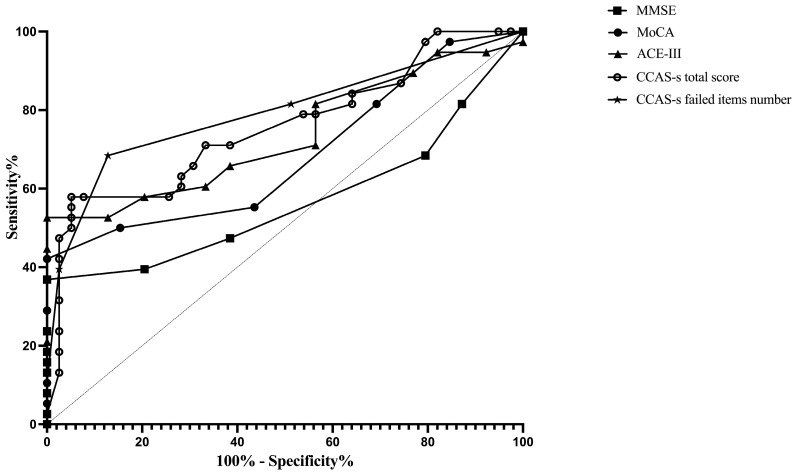
Discriminative ability of MMSE, MoCA, ACE-III, and CCAS-s to identify cognitive impairment in patients with frontal infarctions. Abbreviations: MMSE: Mini-Mental State Examination; MoCA: Montreal Cognitive Assessment; ACE-III: Addenbrooke’s Cognitive Examination III; and CCAS-s: Cerebellar Cognitive Affective Syndrome scale.

**Table 1 brainsci-14-00676-t001:** Demographic and clinical characteristics of patients in cerebellar infarction group, supratentorial infarction group, and healthy group.

	Cerebellar Infarction Patients *n* = 50	Supratentorial Infarction Patients *n* = 60	Healthy Controls *n* = 39	*p* Value
Age (y)	53.1 ± 12.5	50.4 ± 11.9	50.8 ± 7.8	0.34
Male, *n* (%)	39 (78.0)	44 (73.3)	30 (76.9)	0.84
Education (y)	11.1 ± 3.0	11.2 ± 3.0	11.0 ± 2.0	0.94
BMI (kg/m^2^)	23.4 ± 0.4	23.0 ± 0.3	22.7 ± 0.4	0.34
Current smoker (*n*, %)	15 (30.0)	26 (43.3)	12 (30.8)	0.38
Current drinker (*n*, %)	16 (32.0)	17 (28.3)	10 (25.6)	0.51
Medical history (*n*, %)				
Hypertension	34 (68.0)	45 (75.0)	24 (61.5)	0.36
Diabetes	18 (36.0)	20 (33.3)	12 (30.8)	0.06
Dyslipidemia	11 (22.0)	16 (26.7)	15 (38.5)	0.22
Disease duration (d)	8.7 ± 4.9	7.1 ± 3.8	-	0.03
Lesion volume (cm^3^)	17.0 ± 8.1	17.8 ± 8.2	-	0.48
NIHSS score	1.1 ± 0.9	1.2 ± 1.3	-	0.63
ICARS score	7.5 ± 6.9	-	-	-
BBA score	11.0 ± 1.4	-	-	-
SAS score	36.3 ± 5.5	35.7 ± 6.8	31.6 ± 6.0	0.001 ^a,b^
SDS score	35.7 ± 6.7	35.0 ± 6.6	35.6 ± 5.8	0.73

^a^: patients with cerebellar infarctions were significantly different from healthy controls (*p* = 0.001); ^b^: patients with supratentorial infarctions were significantly different from healthy controls (*p* = 0.004). Abbreviations: BMI: body mass index; NIHSS: National Institutes of Health Stroke Scale; ICARS: International Cooperative Ataxia Rating Scale; BBA: Brunel Balance Assessment; SAS: Self-rating Anxiety Scale; and SDS: Self-rating Depression Scale.

**Table 2 brainsci-14-00676-t002:** Global cognitive performance of patients in cerebellar infarction group, supratentorial infarction group, and healthy group.

Screening Test	Cerebellar Infarction Patients *n* = 50	Supratentorial Infarction Patients *n* = 60	Healthy Controls *n* = 39	*p* Value
MMSE	27.7 ± 1.6	25.8 ± 3.7	27.7 ± 1.3	0.05
MoCA	27.5 ± 1.5	26.2 ± 2.4	27.9 ± 1.3	0.001 ^b,c^
ACE-III	89.8 ± 4.7	88.4 ± 4.7	93.3 ± 2.9	<0.001 ^a,b^
CCAS-s	93.1 ± 10.1	89.6 ± 8.8	99.0 ± 6.1	<0.001 ^a,b^
Number of CCAS-s failed items	1.8 ± 1.7	2.3 ± 1.4	0.7 ± 0.8	<0.001 ^a,b^

^a^: patients with cerebellar infarctions were significantly different from healthy controls (*p* < 0.05); ^b^: patients with supratentorial infarctions were significantly different from healthy controls (*p* < 0.05); ^c^: patients with supratentorial infarctions were significantly different from patients with cerebellar infarctions (*p* < 0.05). Abbreviations: MMSE: Mini-Mental State Examination; MoCA: Montreal Cognitive Assessment; ACE-III: Addenbrooke’s Cognitive Examination III; and CCAS-s: Cerebellar Cognitive Affective Syndrome scale.

**Table 3 brainsci-14-00676-t003:** Z-transformed cognitive scores of patients in cerebellar infarction group, supratentorial infarction group, and healthy group.

Cognitive Domains	Cerebellar Infarction Patients *n* = 50		Supratentorial Infarction Patients *n* = 60		Healthy Controls *n* = 39		*p* Value ^a^	Post Hoc Analysis		
	Mean ± SD	Range	Mean ± SD	Range	Mean ± SD	Range		Cerebellar/ Supratentorial	Cerebellar/ Controls	Supratentorial/Controls
Attention	−0.73 ± 1.22	−3.34~2.31	−1.38 ± 0.87	−3.83~0.28	−0.00 ± 0.73	−1.59~1.37	<0.001	0.002	0.003	<0.001
Visuospatial	0.09 ± 1.12	−3.53~0.68	0.09 ± 1.05	−2.47~0.68	−0.00 ± 1.00	−4.05~0.68	0.08	−		
Language	−0.04 ± 1.24	−3.24~2.06	0.10 ± 1.19	−3.24~2.06	−0.01 ± 1.01	−2.06~2.06	0.67	−		
Episodic Memory	−0.28 ± 1.03	−2.82~1.35	−0.47 ± 0.89	−2.46~1.35	0.00 ± 0.79	−1.80~1.78	0.03	0.47	0.58	0.02
Executive Function	−0.63 ± 0.96	−3.56~1.03	−0.65 ± 0.93	−3.42~2.26	0.00 ± 0.51	−1.14~1.16	<0.001	0.002	0.003	<0.001

^a^: Kruskal–Wallis H test. Missing data: attention: 1 cerebellar infarction patient (CI) and 1 supratentorial infarction patient (SI); visuospatial: 2 CIs; language: 2 CIs; episodic memory: 1 CI and 1 SI; and executive function: 1 SI. Abbreviations: SD: standard deviation.

**Table 4 brainsci-14-00676-t004:** The risk of impaired cognitive domain in each patient group compared to healthy controls.

Cognitive Domains	No. of Outcomes (*n*, %)			Relative Odds Ratio (95% Confidence Interval)		
	Cerebellar Infarction Patients *n* = 50	Supratentorial Infarction Patients *n* = 60	Frontal Infarction Patients *n* = 38	Cerebellar Infarction Patients	Supratentorial Infarction Patients	Frontal Infarction Patients
Attention	22 (44.9%)	45 (75.0%)	30 (78.9%)	1.58 (1.20–2.09)	3.49 (2.21–5.49)	4.14 (2.21–7.76)
Visuospatial	7 (14.6%)	10 (16.7%)	6 (15.8%)	1.02 (0.86–1.21)	1.05 (0.89–1.23)	1.04 (0.86–1.24)
Language	9 (18.8%)	7 (11.7%)	4 (10.5%)	1.07 (0.90–1.29)	0.99 (0.85–1.15)	0.97 (0.83–1.15)
Episodic Memory	15 (30.6%)	26 (44.1%)	14 (37.8%)	1.22 (0.97–1.53)	1.51 (1.16–1.97)	1.36 (0.98–1.81)
Executive Function	29 (58.0%)	34 (56.7%)	20 (52.6%)	2.02 (1.42–2.87)	1.95 (1.42–2.69)	1.79 (1.25–2.56)

Missing data: attention: 1 cerebellar infarction patient (CI), 1 supratentorial infarction patient (SI), and 1 frontal infarction patient (FI); visuospatial: 2 CIs; language: 2 CIs; episodic memory: 1 CI, 1 SI, and 1 FI; and executive function: 1 SI.

**Table 5 brainsci-14-00676-t005:** Z-transformed cognitive scores of patients in cerebellar infarction group, frontal infarction group, and healthy group.

Cognitive Domains	Cerebellar Infarction Patients *n* = 50		Frontal Infarction Patients *n* = 38		Healthy Controls *n* = 39		*p* Value ^a^	Post Hoc Analysis		
	Mean ± SD	Range	Mean ± SD	Range	Mean ± SD	Range		Cerebellar/ Frontal	Cerebellar/ Controls	Frontal/ Controls
Attention	−0.73 ± 1.22	−3.34~2.31	−1.36 ± 0.88	−3.73~0.28	−0.00 ± 0.73	−1.59~1.37	<0.001	0.009	0.003	<0.001
Visuospatial	0.09 ± 1.12	−3.53~0.68	0.12 ± 1.05	−2.47~0.68	−0.00 ± 1.00	−4.05~0.68	0.09	−		
Language	−0.04 ± 1.24	−3.24~2.06	0.19 ± 1.23	−3.24~2.06	−0.01 ± 1.01	−2.06~2.06	0.46	−		
Episodic Memory	−0.28 ± 1.03	−2.82~1.35	−0.20 ± 0.90	−1.45~1.35	0.00 ± 0.79	−1.80~1.78	0.41	−		
Executive Function	−0.63 ± 0.96	−3.56~1.03	−0.62 ± 1.03	−3.42~2.26	0.00 ± 0.51	−1.14~1.16	0.001	1.00	0.002	0.004

^a:^ Kruskal–Wallis H test. Missing data: attention: 1 cerebellar infarction patient (CI) and 1 frontal infarction patient (FI); visuospatial: 2 CIs; language: 2 CIs; and episodic memory: 1 CI and 1 FI. Abbreviations: SD: standard deviation.

## Data Availability

The data are not publicly available due to privacy or ethical restrictions.

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
