# Peer review of "A Comparative Study on Cognitive Assessment in Cerebellar and Supratentorial Stroke"

_brainsci, 2024, doi:10.3390/brainsci14070676_

Round 1
Reviewer 1 Report
Comments and Suggestions for Authors
Dear authors,
The manuscript is very interesting, providing novel insights on the cerebellar function. My comments are below:
- The introduction is well-structured, but it lacks more about evidence concerning the cerebellar role on executive functions. I suggest you to explore more on that, especially with novel techniques, as tractography.
- Insert the previous established sensitivity and specificity of all questionnaires.
- How did you control the multiple testing? I strongly suggest you to use a post hoc correction after the Kruskall-Wallis or factorial ANOVA. Also, if you use a log transformation there's a possibility to normalize all data. If so, you may use within-between group ANOVA comparison. Please, consider that.
- I could not notice the a priori sample size calculation. Please, clarify and insert.
- The ROC curve analysis was not clear for me. The sensitivity and specificity, as well as the accuracy were not established, nor a cutoff point. Also, in your graphs, I recommend to sepparate them all. It was a little bit confusing.
- Please, be aware of tables format. Some of them are misformatted.
- The discussion is pertinent and also is the conclusion.
- The supplemental material must be described accordingly.
Author Response
The manuscript is very interesting, providing novel insights on the cerebellar function. My comments are below:
1. The introduction is well-structured, but it lacks more about evidence concerning the cerebellar role on executive functions. I suggest you to explore more on that, especially with novel techniques, as tractography.
Reply: Thanks for the comments. We appreciate your suggestion to explore more evidence regarding the cerebellar role in executive functions, particularly through the use of novel techniques such as tractography. We supplemented the relevant contents in the introduction part (line 41-47, 53-59).
2. Insert the previous established sensitivity and specificity of all questionnaires.
Reply: Thanks for the important suggestion. In response to the comment, considering the length of the article, we have added the previously established sensitivity and specificity of all questionnaires to the supplementary materials (line 123-125, table S1).
3. How did you control the multiple testing? I strongly suggest you to use a post hoc correction after the Kruskall-Wallis or factorial ANOVA. Also, if you use a log transformation there's a possibility to normalize all data. If so, you may use within-between group ANOVA comparison. Please, consider that.
Reply: Thanks for the suggestion. We have already used the Bonferroni correction to control the multiple testing (line 136) and the relevant outcome were showed in the result section (table 1-3 and 5). Your input is greatly appreciated. By controlling for multiple testing, we can maintain the overall error rate at an acceptable level and ensure that findings are truly meaningful and not simply the result of chance.
4. I could not notice the a priori sample size calculation. Please, clarify and insert.
Reply: Thank you for your feedback concerning the calculation of the sample size in our study. In the preliminary stage of our research, we have indeed considered the importance of calculating the sample size. However, due to various constraints and specific characteristics of our study, we decided to proceed without a priori sample size calculation. Our study was exploratory in nature and aimed to generate hypotheses rather than confirm them, which is why we didn't pre-calculate the sample size. In addition, the study's uniqueness and novelty made it difficult for us to estimate the effect size, making the sample size calculation challenging. We acknowledge that not calculating the sample size beforehand may limit the statistical power of our study. However, we believe that our findings provide valuable insights and can serve as a foundation for future research in this area, where a more exact sample size calculation can be formed. We mentioned the limitation in the discussion part (line 362-368).
5. The ROC curve analysis was not clear for me. The sensitivity and specificity, as well as the accuracy were not established, nor a cutoff point. Also, in your graphs, I recommend to sepparate them all. It was a little bit confusing.
Reply: We agree with your opinion. We separated the figure 1 into three figures according different groups (Figure 1-3). Moreover, we investigate the optimal cutoff value based on Youden index and calculate the sensitivity and specificity of each scale in each group (Table S3). The supplemented content was in the result part (line 205-215).
6. Please, be aware of tables format. Some of them are misformatted.
Reply: Thanks for the careful review. We have checked the table again and corrected some of the errors (table 3-4).
7. The discussion is pertinent and also is the conclusion.
Reply: Thank you for acknowledging the relevance of the discussion and conclusion. Your feedback is appreciated.
8. The supplemental material must be described accordingly.
Reply: Thank you for your feedback. We sincerely apologized for not uploading the supplementary materials successfully. The supplemental material is now added in the end of the manuscript (line 529-614).
Reviewer 2 Report
Comments and Suggestions for Authors
Dear authors,
I congratulate you on the quality of your article. It is particularly meticulous in its methodology and expression. The data you present are especially interesting. Please allow me to make a few comments that I would like to share with you:
Commnent 1: Maybe you could add: “This study was conducted in compliance with the ethical principles outlined in the Declaration of Helsinki." –line 82
Comment 2: In your study, you did not restrict your patient selection to a specific age group. Could this age variation potentially bias your results? Specifically, individuals over 65 years old, and even some younger individuals, might have had pre-existing cognitive impairments prior to their stroke. How did you ensure that such pre-existing conditions did not confound your findings?
Thank you very much
Author Response
I congratulate you on the quality of your article. It is particularly meticulous in its methodology and expression. The data you present are especially interesting. Please allow me to make a few comments that I would like to share with you:
1. Maybe you could add: “This study was conducted in compliance with the ethical principles outlined in the Declaration of Helsinki." –line 82
Reply: Thank you for the thorough review. We have placed the sentence you mentioned in the appropriate location (line 101-102).
2. In your study, you did not restrict your patient selection to a specific age group. Could this age variation potentially bias your results? Specifically, individuals over 65 years old, and even some younger individuals, might have had pre-existing cognitive impairments prior to their stroke. How did you ensure that such pre-existing conditions did not confound your findings?
Reply: Thank you for bringing up this important point. The potential for pre-existing cognitive impairments in older individuals, as well as in some younger patients, is a valid concern that could impact the results of our study. To minimize this bias, we used the Information Questionnaire on Cognitive Decline in the Elderly (IQCODE) to rule out premorbid cognitive impairment for each participant (line 115-116). Additionally, the healthy controls we recruited were matched in terms of age and education with the patient group. And the statistical results showed no significant different among the three groups (line 144-146, table 1). By adjusting the variables of age and education and comparing cognitive changes between stroke and control groups, we aimed to isolate the effects of stroke on cognitive function. Longitudinal studies are needed in the future to exclude the influence of pre-existing conditions on the results and provide more precise insights into cognitive outcomes following cerebellar and supratentorial stroke (line 368-371).
Reviewer 3 Report
Comments and Suggestions for Authors
Dear Authors
I had the pleasure of reading the manuscript entitled "Cognitive Assessment in Cerebellar Stroke: A Comparative Study with Supratentorial stroke". The article is interesting but there is space for further improvement.
1. Why did the authors decided to include a selective group of patients - NIHHS < 4?
2. Did the authors estimate the required sample size before participants's enrollment?
3. Were the subjects consecutively enrolled?
4. How did the authors select the healthy controls?
5. When exactly were the subjects enrolled (eg. in the acute phase?) and when exactly were the assessments performed?
6. Do the authors have any information on stroke etiology?
7. Do the authors have any information on comorbidities?
8. Did the authors include any subject with more than one past stroke?
8. Authors did not assess several author confounding factors that may have influenced the outcomes, even in minor stroke, such as fatigue. This must be acknowledged, discussed and properly quoted (see doi: 10.1016/j.jstrokecerebrovasdis.2021.105964).
Author Response
I had the pleasure of reading the manuscript entitled "Cognitive Assessment in Cerebellar Stroke: A Comparative Study with Supratentorial stroke". The article is interesting but there is space for further improvement.
1. Why did the authors decided to include a selective group of patients - NIHHS < 4?
Reply: We felt sorry for the lack of explanation about the reason why we selected patients whose National Institutes of Health Stroke Scale (NIHSS) ≤ 3. NIHSS score of ≤ 3 is the criteria for diagnosing minor stroke (DOI: 10.1161/strokeaha.109.572883). (line 88-92) We selected patients who got minor stroke in order to reduce the potential confounding effects of severe stroke symptoms (e.g. severe limb weakness or dysarthria) on cognitive evaluation. This selection strategy allowed us to investigate the cognitive profiles in a relatively homogeneous group of stroke patients with mild symptoms, which make the result more accuracy and reliable.
2. Did the authors estimate the required sample size before participants's enrollment?
Reply: Thanks for the important suggestion. Both You and Reviewer 1 have the concern about the priori sample size calculation. During the initial phase of our research, we did consider the significance of determining the sample size. Nevertheless, due to various constraints and the unique nature of our study, we made the decision to proceed without conducting an a priori sample size calculation. Our study was exploratory in essence, focusing on hypothesis generation rather than validation, hence the absence of pre-calculated sample size. Furthermore, the distinctive and innovative aspects of our study posed challenges in estimating the effect size, complicating the sample size determination. While we acknowledge that not determining the sample size in advance may compromise the statistical power of our study, we believe that our findings offer valuable insights and can lay the groundwork for future research in this field, where a more precise sample size calculation can be established. We have addressed this limitation in the discussion section (line 362-368).
3. Were the subjects consecutively enrolled?
Reply: Yes, the subjects were consecutively enrolled, we modified the description about the participant recruitment (line 82-84).
4. How did the authors select the healthy controls?
Reply: We recruited healthy controls through community advertisements and medical examination department. And all healthy participants underwent medical examinations to they had no brain disorders, psychiatric condition and psychotropic drugs which affecting cognitive functions. We added this in the part of participant recruitment (line 92-94, 96-98).
5. When exactly were the subjects enrolled (eg. in the acute phase?) and when exactly were the assessments performed?
Reply: Thanks for the careful review. Inclusion criteria for patients involved enrollment during the acute phase, with assessments conducted within 14 days of disease onset. We supplemented these in the method part (line 88-92, 114-115).
6. Do the authors have any information on stroke etiology?
Reply: Thank you for your query. Unfortunately, we do not have specific information about the stroke etiology in the dataset that was used for this study. This is indeed a limitation of our study and we have now mentioned this in the limitations section of our manuscript (line 357-360).
7. Do the authors have any information on comorbidities?
Reply: Thanks for the suggestion. We supplemented the medical histories of diabetes mellitus, dyslipidemia, hypertension, and the habit of smoking and drinking. There is no significant difference for the above variables among three groups (line 104-106, 144-146, table 1 and table S2).
8. Did the authors include any subject with more than one past stroke?
Reply: We apologize for the misunderstanding caused by the unclear description of the inclusion criteria. All patients suffered stroke first time. We clarified it in the inclusion criteria (line 88-92).
9. Authors did not assess several author confounding factors that may have influenced the outcomes, even in minor stroke, such as fatigue. This must be acknowledged, discussed and properly quoted (see doi: 10.1016/j.jstrokecerebrovasdis.2021.105964).
Reply: Thanks for the kind suggestion. We acknowledge the potential influence of unmeasured confounding factors, such as fatigue and the etiology of the stroke, on our study outcomes. We agree that failure to control for these factors may have introduced bias to our results. In the future, we will strive to better consider and assess these potentially confounding factors in our research. We have adjusted our discussion to acknowledge this limitation and have also referred to relevant literature in the field (line 357-360). Thanks for bringing this to our attention.
Reviewer 4 Report
Comments and Suggestions for Authors
Dear Authors,
I have read your manuscript, which aimed to understand the cognitive profiles of cerebellar infarction patients and compare them to those with supratentorial infarction, particularly frontal infarction. Also it aims to find reliable assessment tools for detecting cognitive impairment in cerebellar infarction patients.
The topic of the paper has clinical significance. However, ithenticate report showed 40% match. This needs to be revised. Large percent of this is auto-plagiarized from your previous study with similar topic. Please make the necessary changes.
Title should be changed. You only investigated frontal lobe stroke so why mention supratentorial stroke in the title and in the manuscript. Please make the change.
Methods section lacks more data on stroke diagnosis. Which stroke you evaluated – acute, subacute, chronic? Which sequences in MRI were used and on which maschine?
Statistical analysis is poor. Results and Discussion section are adequately described.
Manuscript is overall good. The sample size is unfortunately small, given the pathology.
I recommend revision.
Comments on the Quality of English Language
Minor editing
Author Response
I have read your manuscript, which aimed to understand the cognitive profiles of cerebellar infarction patients and compare them to those with supratentorial infarction, particularly frontal infarction. Also it aims to find reliable assessment tools for detecting cognitive impairment in cerebellar infarction patients.
1. The topic of the paper has clinical significance. However, ithenticate report showed 40% match. This needs to be revised. Large percent of this is auto-plagiarized from your previous study with similar topic. Please make the necessary changes.
Reply: Thank you for your feedback. We apologize for any misunderstanding. We assure you that our intent was not to plagiarize but to build upon our previous research. We also noticed that the plagiarism checking of articles included the references, which is not common practice. This inclusion may have contributed to the high duplicate rate. We have contacted the assistant editor but the journal editorial office cannot provide another checking report. We have revised the main text to ensure that it is unique and distinct from our prior works while still properly citing any referenced materials.
2. Title should be changed. You only investigated frontal lobe stroke so why mention supratentorial stroke in the title and in the manuscript. Please make the change.
Reply: We appreciate your feedback. However, the aim of our study was to investigate the cognitive profiles of patients with isolated cerebellar infarction and compare them to those who experienced supratentorial infarction and the cognitive assessment for patients with frontal infarction is the part of sensitivity analysis in the manuscript. Therefore, we feel the title “A Comparative Study on Cognitive Assessment in Cerebellar and Supratentorial Stroke” accurately reflects the whole content of our research (line 2-3).
3. Methods section lacks more data on stroke diagnosis. Which stroke you evaluated – acute, subacute, chronic? Which sequences in MRI were used and on which maschine?
Reply: We apologize for the misunderstanding caused by the unclear description of the inclusion criteria. All patients are initial occurrence of minor stroke, and we evaluated them in the acute phase. We clarified it in the inclusion criteria (line 88-92). The infarction lesion of each patient was identified on the sequences of T2-fluid attenuated inversion recovery (FLAIR) in the 3.0 T magnetic resonance imaging scanner (SIEMENS, Germany) (line 86-88).
4. Statistical analysis is poor. Results and Discussion section are adequately described.
Reply: Thanks for the comment. I agree the statistical analysis could be improved and your suggestions are beneficial. We further did a post hoc correction by using the Bonferroni analyses after the Kruskall-Wallis test to address any potential issues with multiple comparisons, which should enhance the robustness of our results (line 134-136, table 1-3 and 5). In addition, based on the recommendation of reviewer 1, we also added the optimal cutoff value based on Youden index and calculate the sensitivity and specificity of each scale in patients’ group. This should provide deeper insight and augment the interpretability of our findings (line 205-215, table S3). Furthermore, we have gone through the tables and corrected any errors (table 3-4). This will ensure the accuracy and clarity of the data represented. We appreciate your constructive inputs and believe these modifications will significantly improve the manuscript’s quality. We look forward to your further comments).
5. Manuscript is overall good. The sample size is unfortunately small, given the pathology.
Reply: We fully agreed with you opinion. The limitation of the small sample size has been discussed in the discussion part (line 364-366). In future studies, we will strive to include a larger sample size to further validate and broaden these findings.
Round 2
Reviewer 1 Report
Comments and Suggestions for Authors
Dear Authors,
Thank you for your prompt feedback. No further questions.
Reviewer 3 Report
Comments and Suggestions for Authors
No further comment.